# From Dissonance to Insights: Dissecting Disagreements in Rationale Construction for Case Outcome Classification

**Shanshan Xu**[1*]**, Santosh T.Y.S.S**[1*]**,Oana Ichim**[2]**, Isabella Risini**[3]**,**
**Barbara Plank**[4,5]**, Matthias Grabmair**[1]

[1]Technical University of Munich, Germany
[2]Graduate Institute of International and Development Studies, Switzerland
[3]Faculty of Law, Ruhr University Bochum, Germany
[4]IT University of Copenhagen, Denmark
[5]LMU Munich & Munich Center for Machine Learning (MCML), Germany

## Abstract

In legal NLP, Case Outcome Classification (COC) must not only be accurate but also trustworthy and explainable. Existing work in explainable COC has been limited to annotations by a single expert. However, it is well-known that lawyers may disagree in their assessment of case facts. We hence collect a novel dataset **RAVE: Ra**tionale **V**ariation in **ECHR**[1], which is obtained from two experts in the domain of international human rights law, for whom we observe weak agreement. We study their disagreements and build a two-level task-independent taxonomy, supplemented with COC-specific subcategories. We quantitatively assess different taxonomy categories and find that disagreements mainly stem from underspecification of the legal context, which poses challenges given the typically limited granularity and noise in COC metadata. To our knowledge, this is the first work in the legal NLP that focuses on building a taxonomy over human label variation. We further assess the explainablility of state-of-the-art COC models on RAVE and observe limited agreement between models and experts. Overall, our case study reveals hitherto underappreciated complexities in creating benchmark datasets in legal NLP that revolve around identifying aspects of a case's facts supposedly relevant to its outcome.

## 1 Introduction

The task of case outcome classification (COC) involves classifying the outcome of a case from a textual description of its facts. While high classification performance is of course desirable, its greatest potential utility for legal experts lies in the identification of aspects involved in the case that contribute to the outcome prediction. For example, if these extracted aspects could be grouped

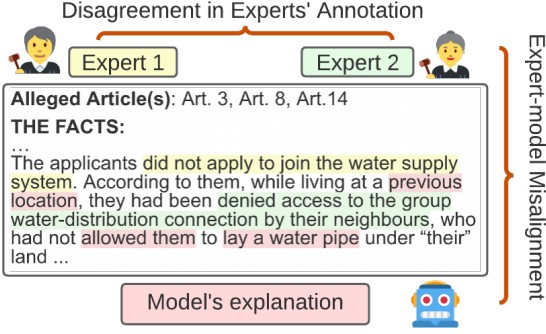

Figure 1: The disagreement between experts' annotation and the misalignment of model and experts'.

into patterns, they can be used to index legal analytics databases that inform litigation strategy. In other words, the case outcome signal can serve as a proxy for legal *relevance* of certain parts of the fact description.

A major obstacle in realizing this vision is the misalignment between what models predict from with what experts consider relevant. For instance, prior work by Chalkidis et al. 2021 demonstrates that the alignment between models and experts rationale is fairly limited in the jurisprudence of the European Court of Human Rights (ECtHR). Previous work (Santosh et al. 2022) shows that these models are drawn to shallow spurious predictors if not deconfounded. Alignment is typically evaluated by deriving a saliency/importance map from the model's predictions on the input text and comparing it with the ground truth labelled by, in most cases, a single expert. Aforementioned previous works have derived and evaluated the explanation rationales at the fairly coarse paragraph level, which may over-estimate alignment.

In light of this, we introduce RAVE, a novel explanation dataset of COC rationales from two legal experts at a much more granular and challenging token level. In addition, while many cases involve multiple allegations, previous works typically col-

---

[1]Our guidelines, dataset and code is available at https://github.com/TUMLegalTech/RaVE_emnlp23

*These authors contributed equally to this work

lect only one annotation per case. In contrast, our approach involves collecting annotations for every alleged article under each case, enabling us to capture a more nuanced decision-making process of the annotators. However, we observed weak inter-annotator agreement (IAA) between experts' rationale annotations, indicating the challenge of marking up rationales consistently and emphasizing the perspectivist nature of the legal decision-making process (see Figure 1).

There is a growing body of work in mainstream NLP highlighting the presence of irreconcilable variations among annotations, and finding that such disagreement is abundant and plausible (Pavlick and Kwiatkowski, 2019; Uma et al., 2021; Sap et al., 2022). Researchers further argue that we should embrace such *Human Label Variation* (HLV) (Plank, 2022) - signal which provides rich information signal, and not as noise which should be discarded (Aroyo and Welty, 2015; de Marneffe et al., 2012; Basile et al., 2021). This motivates us to study the disagreements we encountered between our experts on rationales for COC.

To the best of our knowledge, this is the first work in the legal domain to systematically study HLV in explanation datasets. We propose a two-level task-independent taxonomy of disagreement sources and introduce COC task-specific subcategories. We also quantitatively assess the effects of different subcategories on experts' annotation agreement. The results indicate that disagreements are mainly due to the underspecified information about the legal context in the annotation task, highlighting the importance of developing better annotation guidelines and schema in legal domain. Further, we assess the state-of-the-art COC models on alignment with respect to obtained experts token-level rationales in various settings. Despite differences in COC prediction performance, we observe consistently low alignment scores between models and experts rationales in different settings (Fig 1), suggesting a need for further investigation on how to make models align its focus better with legal expert rationales.

## 2 Related Work

### 2.1 Tasks on ECtHR Corpora

Previous works involving ECtHR corpus has dealt with COC (Aletras et al., 2016; Chalkidis et al., 2019; Valvoda et al., 2023; Santosh et al., 2022, 2023; Tyss et al., 2023; Blas et al., 2023), argu-

ment mining (Mochales and Moens, 2008; Habernal et al., 2023; Poudyal et al., 2019, 2020), event extraction (Filtz et al., 2020; Navas-Loro and Rodriguez-Doncel, 2022) and vulnerability type classification (Xu et al., 2023). In this work, we focus mainly on COC task using ECtHR corpus studying the sources of disagreement/HLV in COC rationale, to contribute to the field of explainable legal NLP.

### 2.2 Explainable COC

Prior work refers to COC as Legal Judgment Prediction (LJP). Under that term, it has gained wide attention in various jurisdictions (e.g., Aletras et al. 2016, Katz et al. 2017, Yue et al. 2021). In the legal domain, explainability is of crucial importance to make models trustworthy. Chalkidis et al. 2021 investigated the explainability of ECtHR allegation prediction on an annotated dataset of 50 cases by identifying the allegation-relevant paragraphs of judgment facts sections. Also working at the paragraph-level, Santosh et al. 2022 used deconfounding to improve model alignment with expert rationales. Malik et al. 2021 introduced an Indian jurisdiction LJP corpus with 56 cases annotated with explanations at sentence-level by experts. Different from the above works, RaVE contains rationales at the more granular word-level for each case-article pair and systematically investigates the possible sources of disagreement between experts.

### 2.3 Annotation Disagreement

Annotation disagreement has been found in a wide range of NLP tasks, especially not only in highly subjective tasks such as natural language inference (NLI) (Pavlick and Kwiatkowski, 2019), toxic language detection (Sap et al., 2022), but also in seemingly objective tasks such as part-of-speech tagging (Plank et al., 2014). The most similar work to ours is LIVENLI (Jiang et al., 2023), which systematically studies the variations in explanations of NLI task. LIVENLI collects 122 English NLI items, each with at least 10 annotations from crowdworkers. Unlike NLI, our task requires annotators to have deep understanding of ECHR jurisprudence to identify the rationales. Hence, we obtain 76 case-allegation article pairs, each annotated by 2 experts. Despite the lower numbers of annotators, we found low agreement in the rationale annotation, prompting us to study the sources of disagreement systematically.

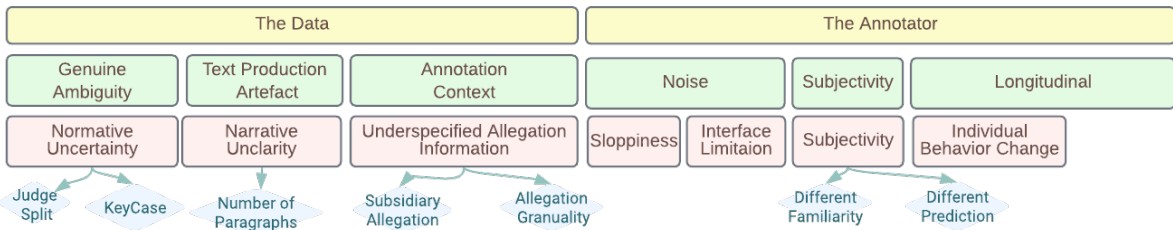

Figure 2: Taxonomy of disagreement sources: Macro Categories (Yellow), Fine-grained Categories (Green), COC rationale annotation specific (Pink), Proxy Variables (Blue).

## 2.4 Categorization of Disagreement

Recently, several studies have proposed taxonomies to identify the possible sources of disagreement in various natural language processing tasks. Basile et al. 2021 outlines three main sources of disagreement: *the annotator* - personal opinions and judgment; *the data* - ambiguity of the text, complexity of the task and *the context* - changes in the subjects' mental state over time. Uma et al. 2021 expanded the list by adding the categories such as: *annotator's error* and *imprecise annotation scheme*. Jiang and Marneffe 2022 introduced a task-specific taxonomy for NLI, consisting of 10 categories that which can be broadly classified into three overarching classes. More recently, Sandri et al. 2023 introduced a taxonomy of disagreement in offensive language detection, which consists of 13 subcategories grouped in four main categories. As pointed out by Sandri et al. 2023, there are many high-level overlaps between these taxonomies, whereas the lower categories, are rather task-specific. Building on the meta-analysis of the existing taxonomies, we generalize the taxonomies of disagreement to a two-layered categories and introduce task-specific categories and proxy variables for COC rationale annotation as in Fig 2. Additionally, we map and unify existing taxonomies into our proposed common categorization, providing an overview of the existing landscape (See Table 4 in App A).

## 3 Ecological Validity

"Ecological validity" refers to the extent to which the experimental situation is similar to real-world condition (Aronson and Lindzey, 1968). At the ECtHR, the application process begins with the applicants lodging their accusation, alleging one or more violations of articles in the ECHR. This process corresponds to Task B (allegation prediction) in the widely used LexGLUE benchmark (Chalkidis et al.,

2022). Subsequently, the court reviews the case and determines whether a violation has occurred, aligning with Task A (violation prediction) in LexGLUE. When ruling on a case, the ECtHR typically examines only the specific articles alleged by the applicant, which corresponds to Task A|B (violation prediction given allegation) as introduced by Santosh et al. 2022. In this study, we evaluate the explainability of COC models for Task A|B, as it mimics the real legal process. We also incorporate ecological validity into the construction of our rationale dataset. Inspired by Jiang et al. 2023, we define a rationale explanation in COC as ecologically valid if the annotator (1) provides both an explanation and a case outcome prediction and (2) to each specific article in allegation. Previous legal rationale datasets, such as Chalkidis et al. 2021 and Santosh et al. 2022, required experts to highlight rationales in the case facts that support the 'violation' of 'any' of the alleged articles. However, we consider this approach not ecologically valid because: Firstly, experts may predict contrary to the given ground truth and thus exhibit bias in explanations when they disagree with the givenn violation label (Wiegreffe et al. 2022; Jiang et al. 2023). Thus, we disclose the allegedly violated articles to the experts, ask them to predict the outcome, and highlight rationales to support their prediction. Secondly, since many cases in the ECHR involve multiple alleged articles, rationales in previous works could justify one or more alleged articles collectively. In contrast, RAVE contains rationales for each alleged article in each case separately, thereby reflecting experts' nuanced decision-making process.

## 4 Dataset Creation

### 4.1 Case Sampling

We sampled 50 cases from the ECtHR's public database HUDOC[*] between 2019–2022. Refer to App B for our sampling criterion. Out of the 50 cases, we allocate 10 cases for pilot study which are used to co-ordinate between experts and refine the guidelines. The remaining 40 were kept as our test set to measure agreement, which contain a total of 76 case-alleged article pairs (hereafter *pairs*).

### 4.2 Annotation Procedure

The rationale annotation process was done using the GLOSS annotation tool (App E) (Savelka and Ashley, 2018). Annotators[*] were given the fact statements of the case, along with information about the alleged article(s). For each alleged article, they were asked to highlight relevant text segments of case fact, which increase the likelihood of them reaching the prediction that a certain article has been violated or not. The full Annotation Guidelines are available in App F. We allow the annotators to highlight the same text span as rationales for multiple articles alleged under that case. We also ask the experts to record their predictions for the outcome of each *pair*, and mark cases they were already familiar with.

**Inconsistent Allegation Information in HUDOC**
During the pilot annotation phase, experts reported occasional incompleteness of the provided allegation information, which we fetched from the HUDOC database. As a result, we recollected the allegation information semi-manually from the case judgment documents for our dataset. For detailed information on this collection process, please refer to App C. We manually inspected all our 50 sampled cases, and identified that such inconsistency exists in approximately 25% of them. Our ECHR experts posit that this issue stems from the fact that the HUDOC primarily registers only the main allegation in the metadata while omitting the subsidiary allegation. As a result, there is a mismatch in the allegation information obtained from HUDOC metadata and perceived by the experts from case text. While it was possible for us to manually curate and correct the metadata involved in this case study, our finding of this HUDOC metadata inconsistency has wider implications for benchmarking efforts derived from the raw database.

---

[*]https://hudoc.echr.coe.int
[*]See App D for annotators' background and expertise

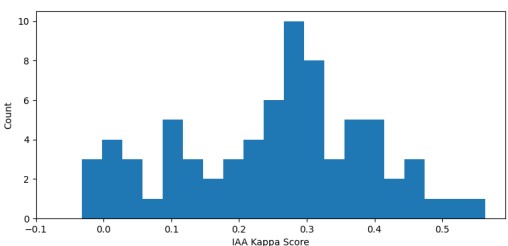

Figure 3: Distribution of the IAA Kappa Scores

### 4.3 The Rationale Dataset and IAA

Table 1 presents the key statistics of our rationale dataset. The dataset comprises 40 cases, with an average of 2764 words per case. On average, each case involves 1.9 allegations, resulting in a total of 76 case-allegation *pairs*. Among these 76 pairs, annotator 1 did not annotate 6 pairs, as she did not find any relevant segment in the case facts for the paired article. Similarly, annotator 2 did not annotate 3 pairs, which were a subset of the aforementioned 6 cases. Annotator 1 tended to provide longer rationales, averaging 485 words per pair; while annotator 2 has average of 373 words per pair. Aligning with a shorter annotation is more challenging as it necessitates capturing the critical aspects of the case text with greater precision.

We measure IAA using Cohen's kappa coefficient $\kappa$ (Cohen, 1960) between the two experts' annotated token-level markups for every pair. We cast expert annotated markup for every pair into a binary vector with of length equal to the number of words in a given case, with a value of 1 if the word is marked up by in annotator and is not an NLTK (Bird et al., 2009) stop word, or 0 otherwise. Figure 3 displays the distribution of IAA kappa scores over the 76 alleged article-case pairs falling in the range of [-0.03, 0.56], with a mean of 0.24. There are 3 out of 76 pairs whose score fall in the negative agreement category, indicating no agreement; 22 out 76 fall into [0.0,0.2], indicating slight agreement; 38 cases have socres between [0.2,0.4], indicating fair agreement, and only 13 scores fall into [0.4,0.6], which deemed to be moderate agreement.

## 5 Disagreement between Experts

### 5.1 Disagreement Taxonomy

To understand the reasons behind experts' disagreement, we present a taxonomy of disagreement sources. The task-independent two-level taxonomy is constructed through a meta-analysis of prior stud-

| Case fact | |
|---|---:|
| # cases | 40 |
| Avg. # allegations per case | 1.9 |
| # case-allegation pair | 76 |
| Avg. length per case | 2764 ±1770 words |
| Avg. # paragraph per case | 40 ± 24 paragraphs |
| Rationales from annotator 1 | |
| Avg. length case-allegation pair | 485 ± 398 words |
| Rationales from annotator 2 | |
| Avg. length case-allegation pair | 373 ± 331words |

Table 1: Statistics for the dataset.

ies on disagreement categorization across different tasks. Additionally, we propose subcategories specific to COC rationale annotation developed together with the two legal experts. We also select proxy variables to quantitatively study the effects of these categories on experts IAA score. The taxonomy was developed by the first author who holds a master's degree in linguistics with expertise in linguistic typology and experience in NLP projects on ECtHR data. To ensure the appropriateness of the proposed subcategories, a discussion and adjudication phase was conducted with the experts to refine and adjust items. Fig 2 offers an overview of the taxonomy and the proxy variables. In the following sections, we provide detailed explanations for each category.

### 5.1.1 The Data

Disagreements within NLP annotation tasks can arise from various aspects of the textual data, which include genuine ambiguity within the text itself, potential artifacts introduced during the text production process or incomplete information within the annotation context.

**Genuine Ambiguity** arises when an expression (a word, phrase, or sentence) can be interpreted in multiple ways. This ambiguity can stem from various language features, such as homonyms and figurative language. Categories from prior works, such as the *Ambiguity* in Uma et al. 2021 and Sandri et al. 2023, and the *Uncertainty in sentence meaning* in Jiang and Marneffe 2022 can be mapped to this category. However, it is important to note that the subcategories are specific to particular tasks. For instance, in NLI, Jiang and Marneffe primarily focus on semantic ambiguity, including subcategories such as *Lexical* and *Implicatures*. Alternatively, in offensive language detection, Sandri et al. emphasizes pragmatic ambiguity, encompass-

ing subcategories such as *Rhetorical Questions* and *Sarcasm*. In COC rationale annotation, genuine ambiguity can be traced back to **Normative Uncertainty**, which arises when multiple legal source interpretation and argumentation is possible to justify an outcome by the court. Its occurrence is not uncommon in ECtHR judgments due to the deliberate drafting of its convention in a relatively abstract manner, to allow interpretation adaptable to a wide range of scenarios.

**Text Production Artefacts** pertain to inconsistencies, incompleteness or introduced biases during text production. The only one category relevant in previous work is *Not Complete* in Sandri et al., which refers to incomplete texts resulting from errors in extracting social media threads. However, in the legal domain, where texts are often lengthy and complex, there is a greater possibility of encountering artefacts during the text production process. Additionally, the facts section of an ECtHR judgment is not an entirely objective account, but the product of registrars' subjective summarization of framing the facts based on the information presented before the Court and their interpretation. Further, case document production often involves multiple court registrars, each may have different narrative style for framing the cases. This can give rise to artefacts such as redundant or inconsistent framing of facts. Consequently, experts may encounter difficulties in interpreting the case due to such **Narrative Unclarity**.

**Context Sensitivity** encompasses instances where annotators may disagree due to a lack of context or additional information needed in the annotation guidelines to interpret a text unambiguously. The *Missing Information* category in Sandri et al. 2023, *Underspecification in Guidelines* in Jiang and Marneffe 2022, and *Item difficulty and Annotation Scheme* in Uma et al. 2021 relate to this category. In our task, the context category arises from **Underspecified Allegation Information**. In comparison to previous studies that only provided experts with factual text, we incorporate additional information regarding the alleged articles. However, feedback from the experts highlights their requirement for more specific allegation information, particularly in cases involving multiple allegation articles. Specifically, they have identified two types of missing information that significantly impact their interpretation of the case: (1) Allegation approach: The distinction between main and

subsidiary allegations has a substantial influence on their comprehension and analysis of the case. (2) Allegation granularity: A more detailed allegation scheme is necessary. For instance, Article 6 of the ECHR comprises multiple sub-parts. Merely annotating for Article 6 as a whole does not enable experts to provide precise and optimal rationale annotations.

### 5.1.2 The Annotator

Disagreements can arise due to variances in annotators' behavior. We consider following categories: **Noise** covers errors due to annotator's Sloppy Annotation or Interface Limitation. **Sloppy Annotation** may occur frequently in crowd-sourcing platforms, where annotators may be recruited without proper training and their annotation quality may not be monitored. Examples that fall under this category include Uma et al. 2021's *Error* and Sandri et al. 2023's *Sloppy Annotation*. Because our annotation process involved close collaboration with legal experts, we expect negligence-related noise to be minimal. While the annotators were actively involved in producing the guidelines during the pilot study, in the main annotation phase these guidelines were tested by new fact patterns and questions about how relevance was to be annotated in individual cases. Limitations of coverage in our guidelines hence contribute to the disagreement we observe in this experiment. We plan to improve on this aspect in future work, but note that case facts can be highly specific and a refinement/extension with explicit instructions for writing styles and narrative patterns from development case examples would consume considerable resources. Additionally, Uma et al. 2021 notes that disagreements in annotation may also stem from **Interface limitations** (i.e., where the capture of relevant information is obstructed by interface quirks). In our case, no such limitation of the GLOSS tool have been brought to our attention.

**Subjectivity** Annotation tasks often involve personal opinions and biases, which are shaped by individual experiences that are inherently private. For instance Sap et al. 2022 demonstrated how annotators' demographic identities and attitudes can influence how they label toxicity in text. In the legal domain, experts' interpretations of cases are inevitably influenced by personal biases. Furthermore, variations in training backgrounds and prior familiarity with cases can also lead to different legal interpretations of a case and hence influence

their choice of rationales.

**Longitudinal** consists of disagreements made by the annotator at different times of testing. While Basile et al. 2021's taxonomy is the only one to consider the temporal aspect of disagreement research, our category is distinct from their *Context* category in two ways. First, *Context* includes *attention slips* of annotator, which we categorize as *noise*. Second, we not only consider **Individuals Behavior Change** of annotators, but also the attitude change of the whole human society towards certain phenomena over time, including changes in laws, policies, and cultural norms. For example, racial segregation used to be institutionalized in many countries, but it has since been legally abolished and is now widely condemned as a violation of human rights. The longitudinal aspect is particularly relevant in the legal domain, where case law evolves over time to reflect changing societal attitudes and values (**Case Law Change**). Lawyers must adapt their legal strategies and reasoning relative to this development. Due to the limited scope of this work, we unfortunately cannot include a longitudinal study and leave it for future research.

### 5.2 Proxy Variables

To quantitatively investigate how the agreement of experts' rationale annotation is influenced by different taxonomic categories, we derive proxy variables together with experts. We assess statistical association between these variables and expert's IAA score. We hypothesize that these proxy variables correspond to lower agreement among experts.

**Normative Uncertainty** enables multiple possible legal interpretations. It is a subjective and abstract concept that is challenging to measure directly from text alone. Therefore, we select the following variables as proxy:

*JudgeSplit*: In cases where multiple interpretations are possible, the judges may fail to achieve an agreement and produce a split voting on the outcome, reflecting the intrinsic ambiguity in the case. We fetch the judges voting record from HUDOC and binarize as 1 if it is a non-unanimous voting, and 0 otherwise.

*KeyCase*: The ECtHR annually chooses a set of significant cases, known as "key cases". These often deal with complex and novel legal issues. Given the absence of established legal standards for interpreting them, they often generate controversy and

disagreement among experts. We retrieve relevant information from the HUDOC and categorize a case as 1 if it is listed as key case and 0 otherwise. **Narrative Unclarity** refers to the lack of clarity in the text resulting from artefacts during the text production process. Experts have emphasised in their feedback that an excessive amount of detail and repetitive descriptions of events or redundant events pose challenges in annotation. In ECtHR judgments, case facts are organized into short paragraphs, each generally representing a distinct factual event or aspect. Consequently, we utilize number of paragraphs (*NumPara*) as a proxy for the level of factual detail.

**Incomplete Allegation Information** encompasses disagreement due to the lack of precise allegation information during annotation, such as the allegation approach and coarse allegation granularity. We select the following two proxy variables:

*OmitAlleg* (Omitted Allegation): As stated in Section 4.2, the ECtHR often omits subsidiary allegations in HUDOC metadata. Here we use these omitted allegations - the allegations present in our curated allegation list but absent in the original HUDOC metadata as a proxy for Subsidiary Allegation.

*Article 6*: According to the experts feedback, cases involving Article 6 (Fair Trial), which includes many sub-parts, lead to considerable uncertainty on which of its aspects was alleged, contributing to their low agreement.

**Subjectivity** pertains to the disagreement due to annotators' personal opinions and bias. We select the following two proxy variables for this category:

*DiffPred* (Difference in Outcome Prediction): Experts may have different outcome predictions on the same case, and further choose different rationales to support their predictions;

*DiffFam* (Difference in Case Familiarity): Prior knowledge of a specific case enables an expert to be aware of its detailed legal and factual details, as well as the court's opinions. Consequently, divergent levels of case familiarity can result in different interpretations.

During annotation, we request experts to indicate their outcome prediction and prior familiarity with each case. We binarize the variable as 1 if both experts have the same prediction or familiarity, respectively, and 0 otherwise.

| Proxy | mean IAA | | t-value | p-value |
|---|---|---|---|---|
| | 0 | 1 | | |
| JudgeSplit | 0.218 | 0.344 | 3.096 | 0.002* |
| KeyCase | 0.242 | 0.249 | -0.183 | 0.856 |
| OmitAlleg | 0.278 | 0.086 | -4.800 | 8e-6* |
| Article6 | 0.260 | 0.150 | 2.378 | 0.020* |
| DiffPred | 0.225 | 0.341 | -2.105 | 0.038* |
| DiffFam | 0.239 | 0.260 | -0.454 | 0.651 |
| NumPara | | | r = -0.269 | 0.019* |

Table 2: Associations of proxy varaibles and IAA scores. (∗: p < 0.05)

## 5.3 Results and Discussion

For each binary proxy variables, we compute the mean of the IAA scores among the group of instances exhibiting that proxy variable (value 1) and the rest (value 0). Then, we perform a independent t-test to compare the mean IAA scores between these two groups. For the ordinal variable *NumPara*, we calculated Pearson's correlation coefficient $r$. The results are presented in Table 2.

Our results show that experts have significantly lower agreement in their annotation for *OmitAlleg* and *Article 6*. This indicates that underspecification in allegation information leads to uncertainty in their annotations, validating our hypothesis which is further confirmed by experts' feedback. We also find significant association between the experts' IAA and *NumPara*. This indicates that the level of detail in the case facts adds to the difficulty. Surprisingly, we found that experts have higher agreement on *DiffPred*. According to experts' feedback, they might have selected the same relevant facts to consider, but their interpretations differ, which can lead to different predictions about the outcome. This was echoed by Jiang et al. 2023 in the context of NLI explanation and entailment labels. Our hypothesis regarding *DiffFam* does not hold and can be attributed to the fact that agreement is more influenced by relative familiarity derived from analogous case fact patterns rather than absolute familiarity with a specific case. When approaching a new case, experts often draw upon their experience with similar cases and apply similar legal principles during annotation. As a result, *diffFam* may not necessarily lead to low agreement. We also observe no significant association for *KeyCase*. This can be attributed to the fact that, while the legal issue might have been complex and novel at the time the case was designated as a Key Case, by the time of annotation, such cases may have transformed into established legal precedents. In

other words, what was uncertain at that time was decided has become settled by now. This temporal change of case law may also explain the counter-intuitive high IAA scores on *JudgeSplit*: the once complex legal issue caused judges split voting in the past may have stabilised by the time of annotation. This highlights the importance of considering the longitude dimension in future studies.

# 6 Expert-Model Misalignment

We evaluate the SOTA COC models' alignment with the experts' rationales in RAVE across two paradigms of models: (i) Fact-only classification models which rely on case facts description as sole input for the violation outcome (Santosh et al., 2022; Chalkidis et al., 2022), and (ii) Article-aware prediction models which take case facts description along with article text to predict the binary outcome of the case with respect to that article (Santosh et al., 2023).

## 6.1 Fact-only Classification Models

As adopted from Santosh et al. 2022, we employ a BERT variant of the hierarchical attention model (Yang et al., 2016), which takes case facts description and the set of articles claimed to be alleged by the applicant as a multi-hot vector as the input and predicts the set of articles that are deemed violated by the court as a multi-hot vector. We use a greedy input packing strategy where we merge multiple paragraphs into one packet until it reaches the maximum of 512 tokens given by BERT models. In this work we experiment with three BERT variants: *BERT* "bert-base-uncased" (Kenton and Toutanova, 2019), *CaselawBERT* "casehold/legalbert" (Zheng et al., 2021), *LegalBERT* "nlpaueb/legal-bert-base-uncased" (Chalkidis et al., 2020). It consists of a pre-trained BERT encoder to obtain token-level representations for every packet. Then these token-level representations are aggregated into packet representation using a token-attention layer. These packet representations are contextualized using a GRU layer and further aggregated into final case fact representation using packet-level attention. Then the final representation is concatenated with the multi-hot allegation vector and passed through two fully connected layers to classify the violation outcome. We train the model using a binary cross-entropy loss against the multi-hot target. See App G for model details.

## 6.2 Article-aware Classification Models

Article-aware prediction for Task A|B takes the case fact text, the text of a convention article, and a binary value indicating whether the given article has been alleged as input. It classifies whether the article has been deemed violated or not by the court, thereby capturing the interplay between case facts and the wording of the convention. We adapt the article-aware architecture from (Santosh et al., 2023) which uses a pre-trained BERT model to obtain pre-interaction token encodings for each token in every packet, which are then aggregated to form packet representations using a token-level attention layer. A bidirectional GRU then contextualizes the packet representations. This procedure is done separately for the text of both facts and article. The interaction component computes the dot product attention between packet representations of case facts and article. We concatenate different combinations of pre-interaction and post-interaction aware representations through concatenation, difference and element-wise product for both article and fact reprsentations. We then merge both branches by conditioning the final representation of case facts on the final article representation obtained from the packet GRU and packet attention layer. This conditioning is done by initializing a bidirectional GRU layer's hidden state of the case facts branch with the post-interaction representation of the article. This guides the GRU model to extract and refine the representations of case facts conditioned on the current article. We concatenate the obtained conditioned case facts representation with the binary label of allegation of that article and use two fully connected layers to obtain the violation outcome. We train the model end-to-end using a binary cross-entropy loss. See App G for model details.

## 6.3 Dataset, & Evaluation Metrics

We use the Task A and Task B datasets from LexGLUE (Chalkidis et al., 2022) and merge them to form Task A|B (i.e., predict convention violations given information about allegedly violated articles) as described in (Santosh et al., 2022). It comprises 11k case fact descriptions chronologically split into train (2001–2016, 9k), validation (2016–2017, 1k) and test sets (2017-2019, 1k). For article-aware classification, we use the expanded dataset from Santosh et al. 2023 with the texts of the 10 articles used in LexGLUE .

We use Integrated Gradient scores to obtain

|  | Model | Explanation Agreement | | Classification Performance | | |
|---|---|---|---|---|---|---|
|  |  | Kappa A1 | Kappa A2 | micro-F1 | macro-F1 | hard-macro-F1 |
| Fact-only | BERT | 11.48 (3.62) | 9.22 (3.29) | 65.52 | 74.97 | 57.97 |
|  | LegalBERT | 11.37 (3.66) | 9.18 (2.94) | 66.03 | 76.49 | 58.42 |
|  | CaselawBERT | 11.59 (3.71) | 9.37 (3.58) | 65.84 | 75.35 | 58.17 |
| Article-aware | BERT | 11.75 (3.64) | 9.57 (3.44) | 74.43 | 80.35 | 61.22 |
|  | LegalBERT | 11.53 (3.51) | 9.31 (3.53) | 74.58 | 80.83 | 61.35 |
|  | CaselawBERT | 11.82 (3.66) | 9.79 (3.22) | 74.49 | 80.95 | 61.38 |

Table 3: Agreement and prediction scores of Fact-only and Article-aware classification. We report Kappa score with standard error for explanation agreement, and F1s for classification performance.

token-level importance from the model with respect of violation outcome of every alleged article (Sundararajan et al., 2017). We max pool over sub-words to convert token-level IG scores into word-level scores, followed by a threshold-based binarization. We report averaged $\kappa$ between the models' focus and the experts' annotations across the case-allegation-article pairs, for each of the two experts. We also report prediction performance in terms of micro-F1 (mic-F1), macro-F1 (mac-F1) and hard-macro-F1 (hmF1), which is the mean F1-score computed for each article where cases with that article having been violated are considered as positive instances, and cases with that article being alleged but not found to have been violated as negative instances following Santosh et al. 2022.

### 6.4 Discussion & Analysis

Table 3 presents classification (F1 scores) and alignment ($\kappa$ scores) performance of fact-only and article-aware classification. We make the following observations: (i) All models across both the settings are slightly better aligned with annotator A1, instead of A2, which may be due to artefact that A1 tends to have longer annotation markup than A2. (ii) Yet overall, agreement between models and human annotators is very low (iii) Though Article-aware models outperform the fact-only variants in all three classification-related metrics, demonstrating the effectiveness of incorporating article information, their alignment with the experts remain similarly low. In other words, *the models might be right for the wrong reasons*. Overall, our results show that there is a need for further investigation and more datasets on how to make models align its focus better with legal expert rationales.

## 7 Conclusion

In this paper we introduce RAVE, a novel corpus for COC annotated with rationale variation provided by multiple legal experts, that accounts for variation in rationales between annotators. We study the human label variation and propose a hierarchical, two-layer taxonomy of sources of disagreement. Our analysis reveals that annotator discrepancies mainly arise from the underspecified allegation information during annotation. We evaluate the interpretability of current state-of-the-art COC models using RAVE and find that there is limited consensus between the models and domain experts. Our results have important implications, highlighting the intricate nature of COC and emphasize the necessity for interdisciplinary collaboration between legal experts and ML researchers to integrate expertise into trustworthy and explainable COC systems that are aligned with legal principles.

## Acknowledgments

We are grateful to Jaromir Savelka for the ability to use the *GLOSS* annotation tool (Savelka and Ashley, 2018). We would also like to thank the members of the MaiNLP/CIS reading group for the inspiring discussions on Human Label Variation. We also thank the anonymous reviewers for valuable comments. BP is funded by ERC Consolidator Grant DIALECT 101043235.

## Limitations

While our two-level taxonomy of disagreement is designed to be task-independent and covers the annotation of various subjective tasks, the specific subcategories introduced in this study are tailored to the COC rationale annotation task. Future investigation is needed to assess the applicability of these subcategories to other tasks within the legal

domain. Similarly, the limited scope of this annotation effort restricts the generalizability of our findings, which remain to be confirmed on a larger dataset with more than two annotators, possibly with different experience profiles. The inclusion of a longitudinal category in Section 5.1 encourages future research to conduct experiments and delve deeper into exploring the temporal aspect within the legal domain.

We focus on in-text rationales for COC, i.e. spans of text in the facts statement marked up by multiple annotators. However, when annotators chose the same rationale elements, they may have had different reasons to do so. Further, in-text rationales only provide evidence for the label without conveying the mechanisms for how the evidence leads to the label. A potentially better way to alleviate both these limitations would be to obtain free-text rationales (Tan, 2022).

Additionally, while our findings of model alignment with experts are specific to the ECtHR domain and datasets, comparable experiments in other domains will see variation based on the nature of the fact input used and the legal issues to be modeled. Nevertheless, all derivation of insight from legal case data comes with modeling assumptions and data-related limitations. We consider this case-study to be prototypical for the kinds of challenges that will have to be tackled to develop future data-driven legal NLP models that are explainable and work in synergy with experts.

## Ethics Statement

Our dataset are retrieved from a publicly available dataset of ECtHR decisions, sourced from the public court database HUDOC . While these decisions include real names and are not anonymized, we do not anticipate any harm arising from our experiments beyond the availability of this information. The task of COC/LJP raises significant ethical and legal concerns, both in a general sense and specifically regarding the European Court of Human Rights (Medvedeva et al., 2020). However, it is important to clarify that we do not advocate for the practical implementation of COC within courts. Previous work (Santosh et al., 2022) has demonstrated that these systems heavily rely on superficial and statistically predictive signals that lack legal relevance. This underscores the potential risks associated with employing predictive systems in critical domains like law, and highlights the im-

portance of trustworthy and explainable legal NLP.

In this study, we focus on obtaining the token-level rationale dataset annotated by experts with an overarching goal to improve the alignment between what models and expert deem relevant. Furthermore, our focus is on investigating the sources of disagreements among lawyers in their assessment of case facts. We have carefully considered the ethical implications of our research and do not anticipate any specific risks associated with it. On the contrary, our analysis of different types of disagreement aims to promote the acceptance of human label variation within the legal NLP community. By acknowledging and understanding various perspectives, interpretations and biases of legal professionals, we contribute to a more comprehensive and inclusive discourse within the field.

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

## A   Disagreement Taxonomy

We map and unify existing taxonomies into our proposed common categorization in Table 4 to offer an overview of the taxonomy landscape.

## B   Case Sampling Criterion

For our rationale dataset, we sampled 50 cases from the ECtHR's public database HUDOC between 2019-22, excluding 'inter-state' cases as suggested by the legal experts. We extract the facts section of the judgement document using regular expressions. Following Chalkidis et al. 2022, we focus on the 10 convention articles that make up the largest share of the court's jurisprudence. We sample 50 cases based on the following criterion: each article should be represented by a minimum of 5 cases in which it is claimed to be violated, and out of which it is deemed to be violated by the court in at least 2 cases, and not found so by the court in at least 2 cases. We use 10 cases as a development set, sampled such that each article is represented by a minimum of 1 case to develop the annotation guidelines.

For the qualitative study, we select five cases based on the following criteria: case which experts have the the highest agreement, case with the lowest agreement, case where two experts have the same outcome prediction but lower rationale agreement, case with different outcome prediction but high in rationale agreement, and case with polarized agreement scores among different allegation articles.

## C   Manual Curation of Allegations

To obtain a corrected set of allegation articles, we first parse the conclusions section in the case judgement document. After noticing that some alleged articles being mentioned in the texts were missing in the conclusion section, to improve quality, we also parsed section headers under the law section in the judgement. We did a manual inspection of all 50 cases sampled for annotation where allegations seemed to mismatch, which amounted to about 25% of the sample. While our test set has been manually curated, we leave an examination of this limitation of HUDOC metadata for future work.

## D   Annotator Background & Expertise

Annotator 1 (the third author) is a Post-doctoral Researcher at a European Research Centre. She worked at the European Court of Human Rights in various roles, including as a case lawyer and a Programme Adviser at the Council of Europe, Directorate General of Human Rights and Legal Affairs. Annotator 2 (the fourth author) is a visiting professor at a European Law Faculty. She

| level 1 | the Data | | | The Annotator | | | | |
|---|---|---|---|---|---|---|---|---|
| level 2 | Genuine ambiguity | Text Production Artefact | Annotation Context | Subjectivity | Noise | | Longitudinal | |
| Basile et al. 2021 | the Data | | | the Annotator | the Context | | | |
| Uma et al. 2021 | | Ambiguity | Item difficulty; annotation scheme | Subjectivity | Error and interface problem | | | |
| Jiang and Marneffe 2022 | | Uncertainty in Sentence Meaning | Underspecification in guidelines | Annotator Behavior | | | | |
| Sandri et al. 2023 | | Ambiguity | Missing information | Subjectivity | | Sloppy annotation | | |
| our work | Normative Uncertainty | Narrative Unclarity | Incomplete Allegation Information | Subjectivity | Interface Limitation | Sloppiness | Individual Behavior Change | Case Law Change |

Table 4: Landscape of existing taxonomies of disagreement sources

spent three months at the European Court of Human Rights as a trainee. Her research interests focus on the supervisory architecture of the European Convention on Human Rights. She served as an expert to the Steering Committee for Human Rights (CDDH) within the Council of Europe.

## E GLOSS Annotation Tool

The task of LJP rationale annotation was done using the GLOSS annotation tool (Savelka and Ashley, 2018). Figure 4 shows a screenshot of the GLOSS annotation interface.

## F Annotation Guidelines

### Task Description: Annotation of Facts Indicating Convention Violations

What we are looking for is to annotate text segments reflecting specific facts that increase the likelihood of the court reaching the conclusion that a certain convention article has been violated or not, respectively. Typically this will be because the fact in question relates to a requirement of a legal rule that is either part of the convention articles, the law as developed in ECtHR's jurisprudence, or other applicable legal source material. The goals of the project include:

- To study to what degree ECHR legal experts agree on what elements of the facts are relevant for the violations of which articles, and

- To develop machine learning models that, given a case fact description, can identify text segments that contain factual information relevant for the determination about whether a given article has been violated. Achieving this will be an incremental step towards building more advanced systems that can support lawyers in analyzing ECtHR jurisprudence and case docket material.

The text to be annotated is the FACTS section of an ECtHR judgment along with information about which convention articles have been alleged to have been violated. The task is to annotate the portions of the text that are relevant for a determination of each alleged article based on this goal. To this end, the GLOSS annotation tool provides one annotation type per article.

### Technical Instructions

Please annotate text spans by clicking and dragging the mouse cursor as you would in a regular text processing environment. Once you release the mouse button, a drop down with the available types will appear for you to select from.

You can annotate the same text with multiple types, but DO NOT annotation the same text with the same type multiple times.

Always annotate the maximal contiguous span. For example: If you want to annotate three consecutive words with type T, then annotate them as one span. DO NOT separately annotate sub-segments of contiguous spans with the same type. Always start annotations at the beginning of a word and end at the end of a word.

**Typical content types to be annotated are:**

- Language segments describing specific facts or fact patterns. Examples: 'They purchased the property with a joint mortgage.', 'She had been interviewed by an investigator from the military prosecutor's office and taken into custody'

- Single- or multiword expressions indicating relevant legal concepts. Examples: margin of appreciation, positive obligation etc.

- Mentions of domestic laws or regulations (e.g. well-known instruments involved in multiple cases) Examples: 'On 3 October 2001 the applicant lodged an application with the Rome Court of Appeal under Law no. 89 of 24 March 2001, known as the "Pinto Act", complaining of the excessive length of the above-described proceedings'

- Specific sequences of events that have a bearing on the finding of a violation or that could justify discarding the applicant's allegation of violation Examples: 'The applicant's allegations were rejected based on a classified note received from the Moldovan secret service, according to which the applicants presented a threat to national security. The decisions did not give any details as to the content of the note, not even the date on which it had been issued' 'However, the court could not see that these measures had had much effect on her ability to provide care.'

- When the ECHR relies in the FACT section extensively on domestic courts' decisions or

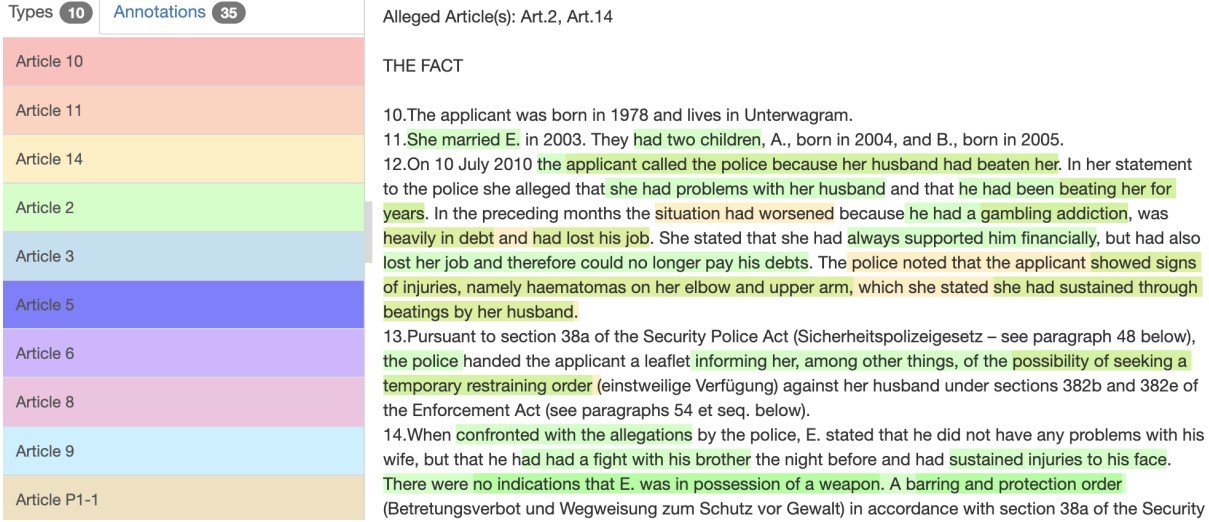

Figure 4: Screenshot of the GLOSS annotation interface

law, we should only annotate facts that are relevant for the alleged violation and not general statements regarding the interpretation of that article in domestic law or even in light of the Convention. We are only looking at facts that would be taken into account into the ECHR's proportionality assessment (the "weighing material" of the national court decision is thus annotated - even if the legal evaluation of that material differs from the national court)

- Specific dates are to be annotated only in cases where those dates disclose events that are relevant for the assessment of the substance of the article Examples: In cases where the arrest is supposed to happen only for a limited time (24 hours or 3 days) and the courts lack the the diligence to review the application or to release the applicant

Focal criterion is the relevance of facts within the particular context of the judgment. Annotation should be conducted by looking through the context to find how facts fit together in order to justify the violation/non-violation. It is the specific composition of facts and laws in the context that makes that violation/non-violation that should be reflected in the annotation. It is important to remember that ECtHR fact descriptions are not objective. They have already been trimmed by the Registry and do not represent the applicant's initial/original allegations, there has also been an exchange between the Government and the applicant and thus the final version on HUDOC is drafted after those facts have been 'set straight'.

**Distinction: What NOT to annotate**
We are not annotating spurious facts, i.e., words, phrases or facts that are not relevant to a finding on the merits regarding the presence or absence of a violation. Redundant/irrelevant facts should not be annotated.

**What to do with repetitive phrases?**
A repetitive phrase occurs when the same fact pattern is written about multiple times in the text. This mainly happens for three reasons, each of which lead to different treatment during annotation: To illustrate the regularity of some event, where one such event would suffice for the legal assessment. In such cases, only one example mention needs to be annotated. We are only looking for facts which reveal the domestic authorities' diligence in dealing with the applicant's allegation (typically the one mentioned first) Example: Cases of prolonged detention in which domestic courts only repeat the initial reasons for arrest without reassessing the applicant's situation Because the presence of repetitiveness is legally decisive, in which case all instances are to be annotated If a fact pattern changes over time or is presented in contradictory perspectives (e.g., differing diagnosis of a mental illness) AND the diversity of information/views will be decisive for the court's assessment (e.g., in determining discharge of positive obligations by the respondent state), then all instances should be annotated. Example:

For example, if the number of meetings might lead to reasonable delay violation (Articles 6, 5

about the promptness of review) or to how many times an investigation was delayed (Articles 2, 3), or how many exams an applicant had to undergo (Articles 8, 3) etc. The underlying rationale is to flesh out only relevant factual patterns and avoid unnecessary and rhetorical language.

**How to annotate special legal vocabulary?**
Given that we are only dealing with the factual part of judgments which contains information of both the respondent Government and of the applicant, terms like "margin of appreciation" and "positive obligation" etc will be annotated only once (see previous rule) and placed in relation with the facts that are relevant for the alleged violation because what we are interested in is to follow how the Court will later use (in dealing with the merits) these rhetorical devices in relation to the facts.

**How to annotate segments relevant for multiple articles?**
In case of conjoint articles (for example 8 and 14, 13 and 8) and in cases where the alleged violations are reflected in the same set of facts (private life and inhumane treatment, property and discrimination) given that the Court might decide for instance that the fact give rise to a violation under the procedural aspect of Article 8 and ignore Article 13, please annotate the same set of facts (with the different colours) for the two articles. This will lead to a slight color change as the markup colorings overlap.

## G  Model Architectures

Fig. 5 illustrates the detailed architecture of both models.

### G.1  Fact-only Classification

We follow the greedy packing strategy and obtain inputs case fact description as $x = \{x_1, x_2, \ldots, x_m\}$ where $x_i = \{x_{i1}, x_{i2}, \ldots, x_{in}\}$. $x_i$, $x_{ij}$, $m$ and $n$ denote the $i^{th}$ packet, the $j^{th}$ token in the $i^{th}$ packet, number of packets, number of tokens in the $i^{th}$ packet, respectively. $b$ and $a$ are provided in form of $\{0, 1\}^k$, where $k$ denotes the total number of modeled convention articles. 1 and 0 indicate a positive and negative instance under allegation (violation) of that corresponding article in $b$ ($a$). [*]

---

[*]We use the terms 'token' and 'packet' instead of 'word' and 'sentence' as used (Yang et al., 2016) as we use BERT as

**Token encoding layer** We use BERT encoders to obtain token-level representations $z_i = \{z_{i1}, z_{i2}, \ldots, z_{in}\}$ for each token in every packet $x_i = \{x_{i1}, x_{i2}, \ldots, x_{in}\}$.

**Token attention layer** We obtain the representation for each packet by aggregating their token-level representations using attention as follows:

$$u_{it} = \tanh(W_w z_{it} + b_w) \qquad (1)$$

$$\alpha_{it} = \frac{\exp(u_{it}u_w)}{\sum_t \exp(u_{it}u_w)} \;\;\&\;\; f_i = \sum_{t=1}^{n} \alpha_{it}z_{it} \quad (2)$$

where $W_w$, $b_w$ and $u_w$ are trainable parameters and $\alpha_{it}$ represents the importance of $t^{th}$ token in the $i^{th}$ packet. Thus we obtain the representations for the input packets as $f = \{f_1, f_2, \ldots, f_m\}$.

**Packet encoding** Given the packet vectors $f$ from token attention layer, we pass them through a bi-directional GRU to obtain context-aware packet representations $g = \{g_1, g_2, \ldots, g_m\}$.

**Packet attention layer** Finally, we aggregate packet representations $g$ into case representation $c$ using attention similar to Eq. 2.

**Classification layer**: We concatenate the multi-hot $b$ with the obtained feature representation $c$ and pass it through two fully connected layers to classify the violation outcome. We again train the model using a binary cross-entropy loss against the multi-hot target.

### G.2  Article-aware Classification

After greedy packing, we obtain case facts description $x = \{x_1, x_2, \ldots, x_m\}$ and article text $s = \{s_1, s_2, \ldots, s_k\}$ where $x_i = \{x_{i1}, x_{i2}, \ldots, x_{in}\}$ and $s_i = \{s_{i1}, s_{i2}, \ldots, s_{ip}\}$. $x_i$ ($s_i$), $x_{ij}$ ($s_{ij}$), m (k) and n (p) denote $i^{th}$ packet, $j^{th}$ token in $i^{th}$ packet, number of packets, number of tokens in $i^{th}$ packet of the case facts description / article respectively. Here $b$ and $a$ are binary variables indicating whether article $s$ has been alleged and violated, respectively.

**Pre-interaction encoding** This layer is applied independently to case facts and article text. We use a pre-trained BERT encoder to obtain token-level representations $z_i = \{z_{i1}, z_{i2}, \ldots, z_{in}\}$ for packet $x_i$. Packet representations $f_i$ are computed via an attention mechanism similar to Eq. 2. Then the packet-representations are passed through a bi-directional GRU to obtain their context-aware

---

backbone which operates on sub-word/token representations. We also merge paragraphs into packets rather than sentences.

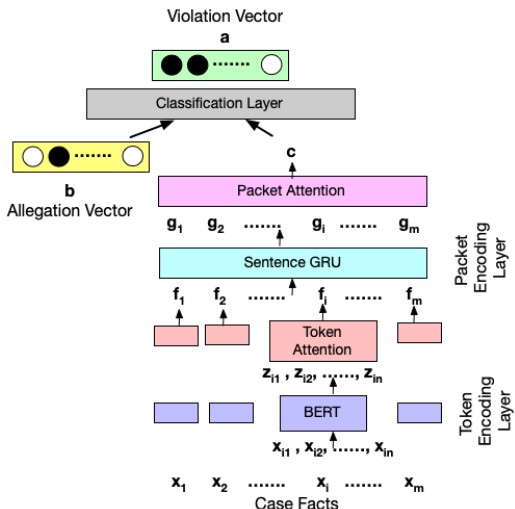

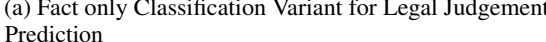

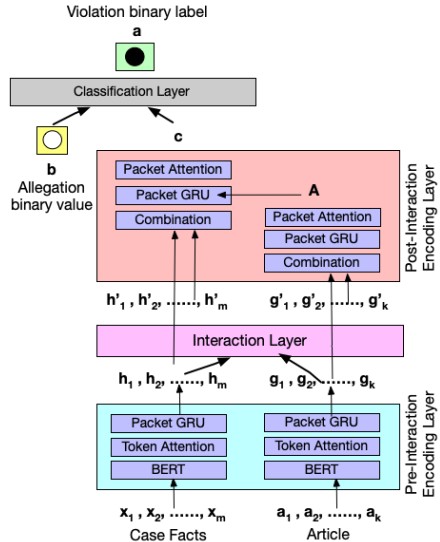

(a) Fact only Classification Variant for Legal Judgement Prediction

(b) Article-aware prediction Variant for Legal Judgement Prediction

Figure 5: Base Model Architectures

representations as $h = \{h_1, h_2, \ldots, h_m\}$. Analogously, we obtain context-aware representations $g = \{g_1, g_2, \ldots, g_k\}$ for the article text.

**Interaction:** This layer captures the interaction between the packets of the case facts description and the article. We compute the article-aware representation of the case facts and fact-aware representations of the article based on relevant content from the packets using the dot product attention mechanism as follows:

$$e_{ij} = h_i^T g_j \quad \& \quad h_i' = \sum_{j=1}^{k} \frac{\exp(e_{ij})}{\sum_{l=1}^{k} \exp(e_{il})} g_j \quad (3)$$

$$g_j' = \sum_{i=1}^{m} \frac{\exp(e_{ij})}{\sum_{l=1}^{m} \exp(e_{lj})} h_i \quad (4)$$

where $e_{ij}$ represents the dot product interaction score between the context-aware representations of the $i^{th}$ packet of case facts and the $j^{th}$ packet of the article. $h_i'$ and $g_j'$ represent the article-aware representation corresponding to the $i^{th}$ packet of the case facts and fact-aware representation corresponding to the $j^{th}$ packet of the article, respectively. Thus we obtain interaction-aware packet representations of the case facts and the article as $h' = \{h_1', h_2', \ldots, h_m'\}$ and $g' = \{g_1', g_2', \ldots, g_k'\}$ respectively.

**Post-interaction Encoding:** This layer uses the interaction-aware representations of both the case-facts and the article to derive the final representation of case facts conditioned on the article text.

Initially, we concatenate pre-interaction and fact-aware packet representations of the article as follows:

$$p_i = [g_i, g_i', g_i - g_i', g_i \odot g_i'] \quad (5)$$

where $\odot$ denotes the element-wise product. This representation is intended to capture higher order interactions between the pre- and post-interaction representations through concatenation, difference and element-wise product. The final packet representations $p = \{p_1, p_2, \ldots, p_k\}$ are passed over a non-linear projection and a bi-directional GRU to obtain the context-aware representations $p' = \{p_1', p_2', \ldots, p_k'\}$. All packet representations are aggregated by an attention mechanism to obtain the final article representation $S$ similar to Eq. 2.

The final fact representations are computed analogously by combining the pre- and post-interaction representations, and then passing them through a non-linear projection and a bi-directional GRU. To condition them on the article text, we initialize the GRU hidden state with the final representation of the given article (Augenstein et al., 2016). This guides the GRU model to extract and refine the representations of case facts not only based on the neighbouring packet contexts but also based on the article text. Then all the packet representations are aggregated to obtain the final representation of case facts $C$ using the same attention computation described in Eq. 2.

**Classification Layer:** We concatenate the above obtained case facts representation $C$ with the bi-

nary label of allegation articles and use two fully connected layers to obtain the violation outcome. We calculate a binary cross-entropy loss over each article in the outcome vector.