# OpenReview forum: "From Dissonance to Insights: Dissecting Disagreements in Rationale Construction for Case Outcome Classification"
_EMNLP/2023/Conference — EMNLP 2023 Main_

### Official Review · Reviewer_CY3m · 2023-08-04

**Typos Grammar Style And Presentation Improvements:** 1. Please present the dataset charact…
**Soundness:** 3

**Excitement:**

2: Mediocre: This paper makes marginal contributions (vs non-contemporaneous work), so I would rather not see it in the conference.

**Paper Topic And Main Contributions:**

The paper creates a new dataset that annotates the facts that lead to a specific legal case outcome .

**Questions For The Authors:**

1. Can u please elaborate on the role of the model performance in Table 2. What is the objective of this table? Is it to show that Article based classification is better than fact only classification?

2. What is the role of  section Dataset, Models & Evaluation Metrics, how are using the LexGlue dataset here. Are u comparing your dataset with LexGlue, this is not entirely clear?

**Reasons To Accept:**

1. Dataset papers are always useful, in addition this dataset contains fine grained annotation of facts that leads to a specific case outcome.

**Reasons To Reject:**

1. The paper presentation can significantly be improved. It is very hard to understand, even though the paper potentially contains useful facts.

**Reproducibility:**

2: Would be hard pressed to reproduce the results. The contribution depends on data that are simply not available outside the author's institution or consortium; not enough details are provided.

**Reviewer Confidence:**

2: Willing to defend my evaluation, but it is fairly likely that I missed some details, didn't understand some central points, or can't be sure about the novelty of the work.

---

> ### Author Rebuttal · Authors · 2023-08-23
>
> We thank the reviewer for the suggestions. We would like to politely clarify that our paper’s contributions go beyond our fine-grained annotation dataset and include the following
> (i) a detailed explainability-to-agreement study and
> (ii) conceptual taxonomy of disagreement sources in the domain of law.
>
> - (3a) Regarding table 2: We will add an additional explanatory table row to make the two insights easier to recognize. The main takeaway from table 2 is although the article-aware model is performing better than the fact-only in overall prediction performance (indicated by f1 scores), this improvement does not bring higher explainability agreement performance as indicated by kappa scores. That means models are predicting better for the wrong reasons and hence they should be further scrutinised. The broader implication for case outcome classification research is that solely relying on prediction performance is not the way forward if the field is to strive for aligning models with experts.
>
> - (3b) Regarding role of LexGLUE: Our agreement annotation dataset extends a subset of cases from the testset of LexGLUE’s ECtHR subtask. We hence evaluate prediction performance and model agreement on those cases by training models on the whole LexGLUE ECtHR task data. In other words, we use the structurally compatible LexGLUE training data to develop models and evaluate them on the annotated and curated test dataset we contribute in this paper.
>
> - (3c) For dataset statistics, please refer to Table 4 in Appendix G where they are provided. We ask understanding that given limited space we use the appendices for general dataset statistics.
>
> - (3d) We thank the reviewer for suggesting a better mnemonic for our task The “A|B” notation is adopted from Santosh et al. 2022. It is highly contextual, though, and we will replace it with a more intuitive name.
>
> - (3e) Model details :
>               - (i) Greedy input packing strategy : we merge multiple paragraphs into one packet until it reaches the maximum of 512 tokens given by BERT models.
>               - (ii) Packet: refers to multiple paragraphs packed into one block of 512 tokens to be sent to bert model
>               - (iii) Contextualization: We follow a hierarchical model where first we obtain each representation of a packet and then those packets are sent through the GRU layer to derive context from neighbouring sentences.
>               - Our model details completely with mathematical equations are contained in Appendix I. Again, we ask for the reader’s understanding of using the appendix for general architectural information.

---

### Official Review · Reviewer_7qoD · 2023-08-05

**Soundness:** 4

**Excitement:**

4: Strong: This paper deepens the understanding of some phenomenon or lowers the barriers to an existing research direction.

**Paper Topic And Main Contributions:**

This paper contributes a novel dataset to the task of Case Outcome Classification. This dataset is built with the involvement of two experts on Human Rights Law. Further, the author/s analyse the disagreement between the experts by introducing a taxonomy augmented with COC-specific terms. The data collected is then tested on reasonable baselines and the author/s observe poor correlation between the model results and the expert annotations.

**Reasons To Accept:**

1. The dataset introduced seems to be a good contribution to the field of explainable-NLP.
2. The annotation process is designed in such a manner that experts would be made to be specific about the impact of specific text to COC, which is a significant benefit.
3. While the author/s do adopt a in-text annotation approach to the task of rationale identification and acknowledge it as a limitation, the approach taken to gauge the impact of certain words and phrases to COC is an interesting one and merits further investigation along the lines proposed.
4. While I have some questions regarding the taxonomy proposed by the authors to gauge annotator disagreement, it is a good contribution and could be helpful in making the subjective nature of COC more objective.

**Reasons To Reject:**

1. The findings of the study seem to indicate more research is needed into understanding why there is disagreement between experts and models in terms of COC. Possibly due to the appreciable difficulty in getting experts to invest significant time and effort into annotating vast legal corpuses, the study involves just two experts annotating the case-allegation pairs, and the disagreement observed between them could be a result of bias inherent in the individual, unrelated to the field's subjectivity. The taxonomy that the author/s propose combines the subjectivity of the field with individual bias which in my opinion we would benefit from decoupling. More experts being involved in the annotation process can also help investigate this claim. [ADDRESSED]
2. In section 4.2, the incompleteness of allegations identified by the author/s in the HUDOC dataset is a significant finding in and of itself, but the fact that it may have resulted in inconsistent information for 25% of cases in the pilot dataset is slightly worrying. Additional information on the corrective measures applied by the author/s may be required in this section. [ADDRESSED]

**Reproducibility:**

4: Could mostly reproduce the results, but there may be some variation because of sample variance or minor variations in their interpretation of the protocol or method.

**Reviewer Confidence:**

3: Pretty sure, but there's a chance I missed something. Although I have a good feel for this area in general, I did not carefully check the paper's details, e.g., the math, experimental design, or novelty.

---

> ### Author Rebuttal · Authors · 2023-08-23
>
> Thank you for your valuable comments.
> - (2a) Difficulty of Getting Experts: We understand your concerns regarding the number of experts. As you mentioned, our legal rational annotation demands deep understanding of ECtHR jurisprudence and extensive reading of legal material, making it very challenging to engage a larger number of domain experts. Nevertheless, in order to uphold the integrity of our proposed taxonomy, we took measures to enhance its quality. We conducted a pilot study, engaged in several rounds of discussions including the experts, and finally also employed an adjudication process in collaboration with the experts.
>
> - (2b) Disagreement Taxonomy: We completely agree with decoupling domain subjectivity and individual annotator bias and have in fact decoupled them by the separate categories “subjectivity” under Annotator (Line 419, representing bias of the individual annotator) and “Normative Uncertainty” under Data (Line 329 capturing with subjectivity of the field - the legal norms).
>
> - (2c) Inconsistent Allegation Information in HUDOC: We appreciate the reviewer agreeing that the HUDOC metadata inconsistencies is an important finding in itself (as prior approaches in this field extensively relied on this metadata). To clarify: We have remedied the problem for data in our experiments, and hence our results are not compromised. We included detailed information on the corrective measures in App C, but will introduce explanations in the main of the paper. In a nutshell, we manually curated/corrected the metadata entries from the text of the decisions, including basing dataset stratification on corrected metadata.

---

### Official Review · Reviewer_UWAY · 2023-08-05

**Soundness:** 4

**Excitement:**

4: Strong: This paper deepens the understanding of some phenomenon or lowers the barriers to an existing research direction.

**Paper Topic And Main Contributions:**

This paper studies the disagreement between human experts when extracting rationales from ECtHR cases to support their predictions of the outcome of the case. Given a case and the allegedly violated articles, the experts are asked to predict the outcome of the case (which articles, if any, of the case were violated) and annotate the tokens of the facts that support their predictions.

The resulting dataset, RAVE, is used to measure the agreement between the experts which is found to be low. To better analyze the disagreements the authors propose a new taxonomy containing potential disagreement reasons. Both the dataset and the taxonomy, along with the guidelines will be made publicly available.

Finally, the authors measure the alignment of SotA models with the rationales of the experts. The authors use two types of models. The first consumes only the facts of the case to predict the allegedly violated articles. The second consumes both the facts and an allegedly violated article and decides if the article was violated. To extract rationales they use Integrated Gradient scores and find that the models do not align with the rationales of the human experts.

**Reasons To Accept:**

* The paper is well-written, clear, and easy to follow.
* The dataset and the taxonomy are useful resources and can be a good first step towards incorporating disagreement when tackling Legal NLP tasks which could potentially lead to improved models.
* The discussion of the disagreement is engaging and gives valuable insights concerning the difficulties of Case Outcome Classification and Legal NLP in general.

**Reasons To Reject:**

* My only concern has to do with the methodology when studying the misalignment between experts and models. In particular, the authors use only Integrated Gradients to extract rationals. On its own, this is not strong evidence that the models are misaligned with the experts. This claim could be better supported by examining more explainability methods including explainability by construction, similarly to Chalkidis et al.

**Reproducibility:**

5: Could easily reproduce the results.

**Reviewer Confidence:**

5: Positive that my evaluation is correct. I read the paper very carefully and I am very familiar with related work.

---

> ### Author Rebuttal · Authors · 2023-08-23
>
> Thank you for your valuable comments.
>
> - (1a) XAI method: We agree that comparing various XAI methods like attention scores or occlusion-based approaches would strengthen the argument, but the reliability and informativeness of these techniques remain an ongoing research challenge. We chose Integrated Gradients due to its prevalent use, notably demonstrated in previous research by Santosh et al. (2022) on ECHR tasks and suitability for token-level evaluation. Occlusion approaches as in Chalkidis et al. are intuitive at paragraph or sentence levels but do not immediately translate to token-level granularity.

---

### Meta-Review · Area_Chair_CoET · 2023-09-18

**Recommendation:** 5

**Metareview:**

The paper examines label variation/human disagreement in case outcome classification.

Reviewers generally agree that the paper presents a valuable contribution, especially as it provides important insight into human subjectivity and limitations of current models. They appreciate the thoughtfulness of the fine-grained dataset and the taxonomy of disagreement sources. One reviewer also notes the methodology as one that opens up opportunities for additional investigation.

---

### Decision · Program_Chairs · 2023-10-07

**Decision:**

Accept-Main

**Comment:**

The paper examines label variation/human disagreement in case outcome classification.

Reviewers generally agree that the paper presents a valuable contribution, especially as it provides important insight into human subjectivity and limitations of current models. They appreciate the thoughtfulness of the fine-grained dataset and the taxonomy of disagreement sources. One reviewer also notes the methodology as one that opens up opportunities for additional investigation.